# An Alpha-Glucan from *Lomentospora prolificans* Mediates Fungal–Host Interaction Signaling through Dectin-1 and Mincle

**DOI:** 10.3390/jof9030291

**Published:** 2023-02-23

**Authors:** Mariana Ingrid Dutra da Silva Xisto, Lucas dos Santos Dias, Francisco Felipe Bezerra, Vera Carolina Bordallo Bittencourt, Rodrigo Rollin-Pinheiro, Ana Carolina Cartágenes-Pinto, Rosa Maria Tavares Haido, Paulo Antônio de Souza Mourão, Eliana Barreto-Bergter

**Affiliations:** 1Laboratório de Química Biológica de Microrganismos, Departamento de Microbiologia Geral, Instituto de Microbiologia Paulo de Góes, Universidade Federal do Rio de Janeiro (UFRJ), Rio de Janeiro 21941-902, Brazil; 2Department of Pediatric, School of Medicine and Public Health, University of Wisconsin-Madison, Madison, WI 53706-1521, USA; 3Laboratório de Tecido Conjuntivo, Hospital Universitário Clementino Fraga Filho and Instituto de Bioquímica Médica Leopoldo de Meis, Universidade Federal do Rio de Janeiro (UFRJ), Rio de Janeiro 21941-913, Brazil; 4Departamento de Microbiologia e Parasitologia, Instituto Biomédico, Universidade Federal do Estado do Rio de Janeiro, Rio de Janeiro 20211-010, Brazil

**Keywords:** *Lomentospora prolificans*, α-glucan, *Scedosporium boydii*, Toll-like receptors, C-type lectin receptors

## Abstract

*Scedosporium* and *Lomentospora* are a group of filamentous fungi with some clinically relevant species causing either localized, invasive, or disseminated infections. Understanding how the host immune response is activated and how fungi interact with the host is crucial for a better management of the infection. In this context, an α-glucan has already been described in *S. boydii*, which plays a role in the inflammatory response. In the present study, an α-glucan has been characterized in *L. prolificans* and was shown to be exposed on the fungal surface. The α-glucan is recognized by peritoneal macrophages and induces oxidative burst in activated phagocytes. Its recognition by macrophages is mediated by receptors that include Dectin-1 and Mincle, but not TLR2 and TLR4. These results contribute to the understanding of how *Scedosporium*’s and *Lomentospora*’s physiopathologies are developed in patients suffering with scedosporiosis and lomentosporiosis.

## 1. Introduction

*Scedosporium* and *Lomentospora* are a group of filamentous fungi comprising some clinically relevant species, such as *Scedosporium boydii*, *Scedosporium apiospermum*, *Scedosporium aurantiacum*, and *Lomentospora prolificans* [1]. The latter was formerly known as *Scedosporium prolificans* but was recently reclassified [2]. *L. prolificans* is one of the most virulent species of the *Scedosporium* and *Lomentospora* group and has therefore been extensively studied [3,4].

*Scedosporium* and *Lomentospora* cause a wide spectrum of diseases in humans. In immunocompetent people, they usually cause a localized infection called mycetoma, resulting from a traumatic inoculation of fungal cells. On the other hand, in immunocompromised patients, including those with HIV/AIDS, organ transplant recipients, and people who suffer of hematological diseases, an invasive and disseminated infection might occur, which is hard to treat and presents high mortality rates [1,3]. 

Activation of the innate immune response is a key step for the clearance of *Scedosporium* and *Lomentospora* infections and to avoid their dissemination [5]. Understanding how these fungi interact with the host and induce the immune response is crucial to improve disease management. In this context, a variety of surface glycoconjugates, such as α-glucans, have already been described in *Scedosporium* and *Lomentospora* species [6]. A glycogen-like α-glucan has been identified in *S. boydii*, which plays a role in inducing the release of pro-inflammatory cytokines by macrophages and dendritic cells, such as TNF-α and IL-12 [7]. The recognition of α-glucan of *S. boydii* involves Toll-like receptor 2, CD14, and MyD88 [7]. 

However, although several glycoconjugates have already been described in *Scedosporium* and *Lomentospora* species, little is known about α-glucan structures and the biological roles in species other than *S. boydii*. In the present work, we describe the structural characterization of an α-glucan from *L. prolificans* and evaluate its role in the induction of innate immune response.

## 2. Materials and Methods

### 2.1. Microorganisms and Growth Conditions

*Lomentospora prolificans* FMR3569 was kindly provided by Dr. J. Guarro from Unitat de Microbiologia, Facultat de Medicina and Institut d’Estudis Avançats, Réus, Spain. Fungal stocks were maintained in solid modified Sabouraud (m-SAB) (2% de glucose, 1% peptone, 0.5% yeast extract, and 2% agar). Cells were inoculated in 200 mL liquid m-SAB and kept for seven days at 25 °C under agitation, after which the mycelium was filtered and washed with distilled water. Conidia were grown on Petri plates containing m-SAB at 25 °C. After 7 days, conidia were obtained after washing the plate surface with sterile phosphate-buffered saline (PBS) by scraping. After filtration, the solution was centrifuged to obtain a pellet containing conidia. Cells were counted using a Neubauer chamber.

### 2.2. Extraction and Purification of α-Glucan

Polysaccharides were isolated from *L. prolificans* mycelia as described previously [7]. Briefly, cells were extracted with 2% KOH solution at 100 °C under reflux for 2 h. The alkali extract was neutralized with glacial acetic acid and centrifuged, and the supernatant containing the polysaccharides was precipitated using three volumes of ethanol. The precipitate was dialyzed against distilled water and lyophilized. The crude polysaccharide was fractioned by gel filtration over a Superdex 200 column using 0.15 M sodium chloride in 0.01 M sodium phosphate, pH 7.0, as an eluant. Eluted fractions were monitored at A 280 nm for protein detection and colorimetrically (A 490 nm) for carbohydrate detection [8]. The purified fractions were submitted to a hydrolysis with 3 M trifluoroacetic acid at 100 °C for 3 h to monosaccharides identification by High Performance Thin Layer Chromatography (HPTLC), using sugar standards [7]. 

### 2.3. Nuclear Magnetic Resonance Spectroscopy (NMR)

α-Glucan spectra were obtained using a 400 MHz spectrometer DRX AVANCE (Bruker). The 13C-NMR (100.6 MHz) spectra were obtained using a solution of α-glucan in 99.7% D_2_O (Bittencourt et al. 2006). Nuclear magnetic resonance (NMR, 1D, and 2D spectra) was performed for polysaccharide fractions (20 mg/0.5 mL D_2_O) using triple resonance probe (US2 900 MHz, Bruker, Karlsruhe, Germany). ^1^H NMR 1D spectra were recorded using 250 scans, with a 1.5 second inter-scan delay at 40 °C. ^1^H-^13^C HSQC (heteronuclear single quantum coherence) spectra were collected using time proportion phase incrementation (TPPI) for quadrature detection in indirect dimension. TOCSY was run with 4046 × 400 points with a spinlock field of 10 kHz and a mixing time of 80 milliseconds. Chemical shifts were displayed relative to external trimethyl-silylpropionic acid at 0 ppm for ^1^H and relative to methanol at 48 ppm for ^13^C [9,10]. 

### 2.4. Production of Rabbit Immune Serum

Immune sera were obtained according to Haido and colleagues [11]. Briefly, male rabbits weighing around 2.5 kg were immunized intramuscularly twice with a one-month interval. The antigen solution consisted of 1 mg/mL lyophilized mycelia in PBS mixed with Complete Freund Adjuvant (1:1). A second boost was given three weeks after the first one. Ten days after the second boost, rabbit blood was collected via cardiac puncture, incubated at 37 °C for 30 min and then overnight at 4 °C. Sera were centrifuged and kept at −20 °C.

### 2.5. Mice

In all experiments female Balb/C mice, 6–12 weeks old, were used. All experiments were performed according to Institutional Committee for Animal Care and Experimentation of the Federal University of Rio de Janeiro (UFRJ), Protocol 027/22/CEUA-UFRJ.

### 2.6. Cytotoxicity of α-Glucan

Mice peritoneal macrophages were added to 96-well plates and incubated in the presence of 3.1, 6.2, 12.5, 25, 50, 100, and 200 µg/m of α-glucan for 24 h at 37 °C with 0.5% CO_2_. After addition of a solution of 0.01%, neutral red cells were incubated at 37 °C for 3 h. Cells were then washed with PBS and fixed with 4% paraformaldehyde for 10 min. Then cells were lysed in a solution of 1% acetic acid and 50% methanol. The number of neutral red cells was determined by spectrophotometer at 492 nm [12,13].

### 2.7. α-Amyloglucosidase Treatment

The treatment with α-amyloglucosidase will remove α-glucan by cleaving off the →4)- αGlc (1→ units.

Conidia treatment: Conidia (1 × 10^8^) were incubated with 6 UI of α-amyloglucosidase in 120 mM acetate buffer, pH 4.2, for 4 h at 56 °C. Enzymatic digestion was stopped by heating the material at 100 °C for 5 min, after which the conidia were thoroughly washed with pyrogenic saline.

Glucan treatment: Glucan (1 mg/mL) was incubated with 6 UI of α-amyloglucosidase in 120 mM acetate buffer, pH 4.2, for 2 h at 56 °C. Then, the mixture was heated at 100 °C for 5 min, to inactivate the enzyme, and centrifuged (3 min/12,000 rpm). The supernatant was concentrated and applied to a HPTLC plate under the same conditions described above [14].

### 2.8. ELISA

*Lomentospora prolificans* α-glucan (25 ng/100 µL/well) or 5 × 10^5^ conidia were added into wells of flat-bottomed polyvinyl microtiter plates (Falcon-Becton & Dickinson, Franklin Lakes, NJ, USA), which were maintained for 1 h at 37 °C and overnight at 4 °C. After washing the plates with 0.05% PBS–Tween 20, the nonspecific sites were blocked by addition of 5% skim milk in 0.1% PBS–Tween 20. Serial dilution of rabbit immune sera against whole *L. prolificans* cells in blocking buffer (100 μL) were added to the wells (dilutions 1/100–1/3200) and incubated for 1 h at 37 °C. Then, antibody binding was measured using goat anti-rabbit IgG antibodies conjugate to horseradish peroxidase (Sigma-Aldrich, MO, USA) at 1/2000 in blocking buffer with incubation for 1 h at 37 °C. Furthermore, O-phenylenediamine was added together with H_2_O_2_ as chromogen. The enzyme reaction was followed by measuring the absorbance at 490 nm using an automated reader (BioRad ELISA Reader, Hercules, CA, USA). For inhibition ELISA test, α-glucan at 25, 50 and 100 µg/mL were separately mixed with rabbit immune sera against whole *L. prolificans* cells (dilution 1/400) in blocking buffer and pre-incubated for 1 h at 37 °C before the incubation with *L. prolificans* conidia coated onto wells. The inhibition ELISA test was performed as described above [15,16].

### 2.9. Flow Cytometer Analysis

Conidia (5 × 10^6^) were treated or not (control) with 6 UI of α-amyloglucosidase for 4 h at 56 °C. Cells were then fixed with 4% paraformaldehyde, washed with PBS, and incubated with 3% bovine albumin for 1 h. Furthermore, samples were incubated with immune sera (1:400) for 1 h. For untreated conidia immune sera were incubated for 30 min with 0, 25, and 50 µg/mL of α-glucan prior to its reaction with conidia. Samples were analyzed using a flow cytometer FACS Calibur (Becton Dickinson, Franklin Lakes, NJ, USA). Data were evaluated using “Windows Multiple Document Interface Flow Cytometry Application (WinMDI) 2.8 version software” [17].

### 2.10. Phagocytic Assay

A solution of 3% thioglycolate (1 mL) was injected into the peritoneum of Balb/C mice to recruit macrophages. Three days after injection, macrophages were collected from the peritoneal cavity with cold RPMI 1640, centrifuged, and counted using a Neubauer chamber. To obtain monolayers, macrophages (5 × 10^5^/well) were plated suspended in RPMI 1640 supplemented with fetal bovine sera (FBS) in 24-well microplates and incubated at 37 °C under a 0.5% CO_2_ atmosphere for 24 h. For the interaction of *L. prolificans* cells and macrophages, conidia (5:1 ratio) were added to macrophage monolayers and kept for 2 h at 37 °C under 0.5% CO_2_. The wells were then washed with PBS and the cells were fixed with Bouin solution and 70% ethanol. After staining with Giemsa, the phagocytic index was determined using an optical microscope [7,13].

In order to evaluate whether α-glucan could inhibit conidia phagocytosis, macrophages were incubated with either 50 or 100 µg/mL of α-glucan for 30 min prior to their interaction with *L. prolificans* cells.

The phagocytic index (PI) was determined by counting a total of 200 macrophages and using the following equation: PI = percentage of macrophages that phagocyted at least one conidium × media of internalized conidia [7,13].

### 2.11. Quantification of Nitric Oxide and Superoxide

Monolayers of macrophages were obtained as described above. Macrophages were incubated with 200 µg/mL of α-glucan treated or not with α-amyloglucosidase and *L. prolificans* conidia also treated or not with α-amyloglucosidase at 37 °C with 0.5% CO_2_. Zymosan was used as a positive control. After 2 h and 18 h of incubation, the supernatant was collected from the wells for the measurement of superoxide and nitric oxide, respectively. Nitric oxide was evaluated by the Griess method reading the absorbance at 540 nm [13], and superoxide production was analyzed by determining the activity of the enzyme superoxide dismutase through MTT reduction assay measured at 570 nm [18].

### 2.12. Cytokines Assays

Monolayers of macrophages were obtained as described above. Polymyxin B (10 μg/mL) was added 5 min prior to the stimulation to avoid LPS contamination. Macrophages were then incubated at 37 °C under 0.5% CO_2_ for 18 h with α-glucan from *L. prolificans* (50, 100, and 200 µg/mL) or *S. boydii* (200 µg/mL) and *L. prolificans* conidia (5:1 ratio), treated or not with α-amyloglucosidase. An LPS (O111:B4, 1 μg/mL) was used as positive control. After incubation, TNF-α concentration was determined by using cytokines kit (BD OptEIA ELISA Set—BD Biosciences, Franklin Lakes, NJ, USA) according to the manufacturer’s instructions.

### 2.13. C-Type Lectin Receptors (CLRs) Reporter Assay

BWZ and B3Z reporter cells expressing Dectin-1, Dectin-2, Dectin-3, or Mincle have been described previously [19]. In a 96-well plate, 10^5^ or 2 × 10^5^ B3Z or BWZ cells/well were incubated for 18 h with 240 ng of *L. prolificans* or *S. boydii* α-glucan. β-galactosidase (lacZ) activity was measured in total cell lysates using CPRG (Roche, Basel, Switzerland) as substrate. OD at 560 nm was measured using 620 nm as reference. Curdlan (5 ug/well), Bl-Eng2 glycoprotein (30 ng/mL), and trehalose-6,6-dibehenate (TDB) (25 ng/mL or 5 ug/well) were used as positive controls for Dectin-1, Dectin-2, and Dectin-3 or Mincle, respectively.

### 2.14. TLR2 and TLR4 Reporter Assay 

HEK-Blue^TM^ hTLR2 and HEK-Blue^TM^ hTLR4 cells (InvivoGen, San Diego, CA, USA) were maintained in growth medium (DMEM, 4.5 g/L glucose, 2 mM L-glutamine, 10% heat-inactivated fetal bovine serum, 100 U/mL penicillin, 100 ug/mL streptomycin, 100 ug/mL normocyn) with 1× HEK-Blue^TM^ Selection (InvivoGen). The parental cell lines HEK-Blue^TM^ Null1 and Null2 (InvivoGen) were used as controls for HEK-Blue^TM^ hTRL2 and HEK-Blue^TM^ hTLR4, respectively. Only cells with 20 passages or less were used. About 5 × 10^4^ cells/well were incubated with 0.5 µg/mL *L. prolificans* or *S. boydii* α-glucan for 18 h at 37 °C; 25 ng/mL Pam3CSK4 (TLR2 agonist) and 10 ng/mL LPS (TLR4 agonist) were used as control. After incubation, 20 µL of cell culture supernatant were incubated with 180 µL of QUANTI-Blue^TM^ solution (InvivoGen) at 37 °C for 4 h. Alkaline phosphatase activity (the reporter gene for HEK-Blue^TM^ hTLR4 and hTLR2) was measured by optical density (OD) measurement at 620–655 nm using a microplate reader. Polymyxin B, a cyclic polypeptide antibiotic, was used to eliminate the effects of endotoxin contamination in some experiments with HEK-Blue^TM^ hTLR4 cells.

### 2.15. Statistical Analysis

All experiments were performed in triplicate, in three independent experimental sets. Statistical analyses were performed using GraphPad Prism version 5.00 for Windows (GraphPad Software, San Diego, CA, USA). One-way ANOVA was used for multiple comparisons using Turkey’s test (Turkey post-test), and a *t*-test was used for two-group comparisons. The 90% or 95% confidence interval was determined in all experiments.

## 3. Results

### 3.1. Isolation and Characterization of L. prolificans α-Glucan

Mycelia of *L. prolificans* were extracted with a 2% KOH solution at 100 °C followed by neutralization with acetic acid and precipitation with ethanol. The crude precipitate was applied to a Superdex 200 column which was eluted with a discontinuous gradient of aqueous NaCl. Two carbohydrate-containing fractions, FA and FB, were obtained (Appendix A). The hydrolysis of these fractions followed by HPTLC examination of the products indicated that FA contained only glucose as a sugar moiety (Appendix A). 

The 1 H NMR spectrum of the purified glucan from *L. prolificans* (Figure 1) showed a similar profile to that of *S. boydii* with few variations (Figure 1). Spectra for both glucans only exhibit signals in the regions of the specific carbohydrate spectra (highlighted in Figure 1) and broad signal profile characteristics of high-molecular-weight compounds, thus indicating that the preparations are free of contaminants.

More specifically, the 1D 1H spectrum of α-glucan from *S. boydii* (Figure 1A) had a set of three anomeric signals at 5.38, 5.33, and 4.98 ppm designated as A, A’, and B, respectively. The B signal in the 1H NMR spectrum of the α-glucan from *L. prolificans* is much lower than that of *S. boydii* and, consequently, the absence of a 5.33 ppm signal (A′). The signals between 4.10 and 3.20 ppm were assigned to ring protons (H2 to H5) and carbon six (H6). In structural studies of complex polysaccharides, 1D 1H NMR show several signal overlaps. Using 2D 1H-13C Heteronuclear Single Quantum Coherence Spectroscopy (Figure 1B), we found that the 1H-13C HSQC profile of the glucan from *S. boydii* is similar to the one found by Bittencourt and colleagues [7], which was described as a glycogen-like polysaccharide consisting of →4)-α Glc(1→ (Unit A; 5.39–100.5 ppm), α Glc(1→ (Unit A′; 5.38–100.1 ppm), and →4,6)- α Glc(1→ (Unit B; 4.98–100.1 ppm). The proposed structures depicted in Figure 1C are based on the chemical shifts obtained in this work with the corresponding published values shown in Table 1.

Likewise, the 1D and 2D 1H-13C HSQC of *L. prolificans* are less complex than that of *S. boydii*, as can be seen in the superposition of the spectra (Figure 1B), with the missing signals corresponding to those assigned to branch units. Therefore, these results indicate that the glucan of *L. prolificans* consists of a predominantly linear polymer with little branching formed essentially by →4)- αGlc (1→ (Unit A) (Figure 1C). 

NMR experiments are often used for quantification because signal intensity is proportional to quantity. In this way, we quantified the anomeric signals of the A+A’ and B units. We used the B unit as a branching scoring and found values very close to the methylation data of 24% for *S. boydii* [7] and 5% for *L. prolificans* (Table 1).

### 3.2. α-Glucan Is Detected on the Surface of L. prolificans Conidia

In order to check the immunogenic role of *L. prolificans* α-glucan, rabbit immune serum against whole *L. prolificans* cells was tested with soluble *L. prolificans* α-glucan by ELISA. This analysis revealed that soluble α-glucan was recognized by rabbit immune serum against *L. prolificans* cells (Figure 2A). Furthermore, an inhibition ELISA test was performed to evaluate the *L. prolificans* α-glucan ability to inhibit the reactivity between the *L. prolificans* intact conidia and rabbit immune serum against *L. prolificans* cells (Figure 2B). The serum reactivity with *L. prolificans* conidia decreased around 60–70% when different concentrations of soluble α-glucan (25, 50, and 100 µg/mL) were used. Consistent with this, flow cytometry showed that serum reactivity with intact conidia was reduced when serum was pre-incubated with soluble α-glucan (Figure 2C). Soluble α-glucan inhibited about 30–40% of serum reactivity against intact conidia in a dose-dependent manner. In addition, the conidia treatment with α-amyloglucosidase, which cleaves off the →4)- αGlc (1→ units, reduced about 40% of sera reactivity with treated conidia (Figure 2C). These results suggest that the α-glucan is accessible on the conidial surface.

### 3.3. α-Glucan Plays a Role in the Phagocytosis of L. prolificans by Macrophages

Microorganisms display on their surfaces a multitude of molecules that have several functions. Some of the molecules might be used by the host to identify the pathogen and initiate a specific response. Figure 3B demonstrates the interaction between *L. prolificans* conidia and peritoneal macrophages after 2 h. To investigate whether α-glucan is involved in phagocytosis of *L. prolificans*, we used peritoneal macrophages as a phagocytic cell model. Pre-incubation of these cells with α-glucan (50 and 100 µg/mL) led to a dose-dependent inhibition of 20–40% of conidia phagocytosis (Figure 3A). To exclude the possibility that this inhibition could be the result of cell damage by α-glucan, a toxicity assay using neutral red reagent was carried out. α-Glucan concentrations up to 200 µg/mL were not toxic for peritoneal macrophages (data not shown). These results suggest that α-glucan present as accessible on the surface of *L. prolificans* and is important for the recognition and internalization of the *L. prolificans* conidia by macrophages (Figure 3).

### 3.4. α-Glucan Induces Nitric Oxide and Superoxide Release by Macrophages

The above results indicate that α-glucan is recognized by macrophages, leading to conidia internalization. To assess the ability of *L. prolificans* conidia or α-glucan to activate microbicidal mechanisms by phagocytic cells, we stimulated peritoneal macrophages with either α-glucan or *L. prolificans* conidia (treated or not with amyloglucosidase) and measured the superoxide and nitric oxide (NO) in the supernatant. Intact conidia were able to induce superoxide and NO production as much as zymosan, whereas conidia treated with amyloglucosidase showed a significant decrease in superoxide and NO production (Figure 4). Soluble α-glucan was also able to induce the production of these reactive oxygen and nitrogen intermediates, while treatment of α-glucan with amyloglucosidase completely abolished their production (Figure 4).

It is important to emphasize that the treatment of conidia with the α-amyloglucosidase did not abolish the production of ROS or even NO, suggesting that other molecules on the surface of *L. prolificans*, such as peptidorhamnomannan or glucosylceramide [13,20], are associated with the microbicidal response activation of these phagocytes.

### 3.5. L. prolificans α-Glucan Is Unable to Induce TNF-α Production

The role of α-glucan from *L. prolificans* conidia in the induction of TNF-α by macrophages was examined. Peritoneal macrophages were stimulated with *L. prolificans* conidia (treated or not with amyloglucosidase) or purified α-glucan, and the supernatant was tested for TNF-α production. *L. prolificans* conidia were able to stimulate TNF-α production, but there was no decrease in TNF-α release when cells were stimulated with conidia treated with amyloglucosidase (Figure 5A). In addition, the stimulation of macrophages with purified α-glucan did not induce TNF-α secretion (Figure 5B), as opposed to what was previously reported by our group for *S. boydii* α-glucan [7].

### 3.6. L. prolificans α-Glucan Is Recognized by TLR2

Peritoneal macrophages stimulated with *S. boydii* α-glucan produce TNF-α in a MyD88- and TLR2-dependent manner [7]. In contrast, stimulation with *L. prolificans* α-glucan does not lead to TNF-α production (Figure 5B). To better understand these findings, we investigated the recognition of both molecules by TLRs. Interestingly, TLR2 reporter cells recognized *S. boydii* α-glucan, corroborating our previous findings [7], but not *L. prolificans* α-glucan (Figure 6A). However, TLR4 did not recognize any of the polysaccharides (Figure 6B). According to a structural analysis of these glucans, the only difference between both is the absence of α-(1→6)-linked glucopyranoside side chains in *L. prolificans* glucan (Figure 1). Our results suggest that such side chains are required for α-glucans recognition by TLR2.

### 3.7. L. prolificans α-Glucan Is Recognized by Dectin-1 and Mincle

In addition to TLR recognition, C-type lectin receptors (CLRs) are crucial to guarantee a proper immune response against fungal infection [21]. To further characterize and understand the difference between α-glucan from *L. prolificans* and *S. boydii* recognition, we used CLR reporter cells to check the recognition of these molecules by Dectin-1, Dectin-2, Dectin-3, and Mincle. We found that *L. prolificans* α-glucan was recognized by Dectin-1 and Mincle but not by Dectin-2 and Dectin-3 (Figure 7). However, *S. boydii* α-glucan was only recognized by Mincle. Together, these findings show for the first time that *L. prolificans* α-glucan is a ligand for Mincle and Dectin-1, mainly through the α-(1→4)-linked glucopyranoside, and that the presence of α-(1→6)-linked glucopyranoside side chains in *S. boydii* α-glucan hinder the recognition by Dectin-1.

## 4. Discussion

Several studies have demonstrated the importance of polysaccharides in the fungal cell, not only contributing to the architecture and integrity of the cell wall, but also as immunologically active components with great potential as regulators of pathogenesis and the host immune response. Among the fungal polysaccharides in *Scedosporium* and *Lomentospora* species, α-glucan and rhamnomannan had their structures characterized and their functions as activators of innate immunity identified [7,22].

α-Glucans have been isolated from a variety of fungal pathogens, and their chemical structures have been characterized. The α-(1→3)-glucan of *Histoplasma capsulatum* masks the recognition of β-(1→3)-glucan by innate immune receptor Dectin-1, which can induce the production of pro-inflammatory cytokines by macrophages [23]. *Aspergillus fumigatus* α-(1→3)-glucan is exposed during conidia germination and induces the aggregation of their germinating conidia [24]. In this work, we describe the structure of an α-glucan from *L. prolificans* to be a linear polysaccharide of (1→4)-linked glucopyranose units, based on a combination of several techniques, including HPTLC chromatography, 1D RMN 1H, and (B) 2D 1H-13C HSQC. These results confirmed that, differently from glycogen-like glucans of *S. boydii* [7], *A. fumigatus* [25], *Mycobacterium bovis* [26], and rabbit liver [27], purified *L. prolificans* α-glucan is a linear polysaccharide.

The reduction in reactivity after α-amyloglucosidase treatment of *L. prolificans* conidia with immune serum against whole fungal cells showed that α-glucan is accessible on the conidia surface to bind with antibodies, thus suggesting that conidia interactions with host cells may be mediated by α-glucan.

The first line of defense against invading microorganisms is the innate immune system, based on the recognition of PAMPs present in pathogens by phagocytic cells, which are important in control the growth and spread of fungi [28,29]. Here we found that the phagocytosis of *L. prolificans* conidia by macrophages relies partially on the recognition of α-glucan. In agreement with these findings, α-glucan from *S. boydii* inhibited conidia phagocytosis by macrophages in a dose-dependent manner. However, the phagocytic receptor involved in the recognition of *S. boydii* α-glucan was not investigated [7].

The recognition of PAMPs by macrophages and neutrophils leads not only to the internalization of the pathogen, but also to the production of microbicidal compounds that damage the invading microorganism. Macrophages and neutrophils produce reactive oxygen species (ROS) through the activation of the NADPH-oxidase complex (NOX), which generates the superoxide anion (O_2_^-^) necessary for the death of microorganisms by producing protein modifications, nucleic acid breaks, and lipid peroxidation [30]. The interaction among macrophages and *L. prolificans* conidia induced an oxidative response, with a high secretion of H_2_O_2_, but more importantly, soluble α-glucan from *L. prolificans* also induced the release of a large amount of H_2_O_2_. However, conidia treated with amyloglucosidase—which removes all the α-glucan from the cell surface—are still capable of inducing a robust H_2_O_2_ production, indicating that *L. prolificans* conidia have other molecules on their surface that activate this microbicidal pathway. Previous results showed that other molecules on the surface of *L. prolificans*, such as PRM or GlcCer [13,20], are associated with the activation of microbicidal responses in phagocytes.

Additionally, the production of reactive nitrogen intermediates by activated macrophages is another important and potent microbicidal strategy. Fungi such as *Penicillium marneffei* [31], *H. capsulatum* [32], *C. neoformans* [33], *C. albicans* [34], and *S. schenckii* [35] have already been described as susceptible to reactive nitrogen intermediates, especially nitric oxide (NO). *L. prolificans* conidia or soluble α-glucan promote NO production, but differently from what we observed for H_2_O_2_ production, conidia treated with amyloglucosidase led to substantially lower NO production. These results highlight the importance of α-glucan in the fungus–macrophages interaction by inducing microbicidal responses.

The role of cytokines in modulating the immune response has been studied to better establish their contribution to the pathogenesis of fungi. Pro-inflammatory cytokines, such as IL-1 and TNF-α, activate macrophages and allow the recruitment of neutrophils in response to microorganisms [36]. TNF-α is a potent activator of endothelial cells which helps in the migration of monocytes and neutrophils to the sites of infection, and it is also required for the clearance of fungal infection [28,37]. *L. prolificans* conidia induced TNF-α release by peritoneal macrophages, which has already been described [13]. However, *L. prolificans* α-glucan does not seem to be involved in the induction mechanism of this cytokine. In contrast, *S. boydii* α-glucan induces TNF-α release by peritoneal macrophages [7]. 

α-Glucans consisting of highly α-(1→6)-branched α-(1→4)-glucan, glycogen-like structures have been reported to possess immunomodulatory activity related to recognition and signaling by TLR2 [7,38]. Here, experiments using TLR2 reporter cells revealed that *L. prolificans* linear α-(1→4)-glucan is not able to trigger TLR2 intracellular signaling. Members of the C-type lectin receptor (CLR) family are predominant fungal-sensing PRRs that elicit several innate immune responses as well as shape adaptive immunity. Mincle is a C-type lectin receptor (CLR) capable of recognizing mycobacterial cord factor (trehalose dimycolate) and its analog trehalose dibehenate, but not mycolate or trehalose alone. A bipolar glycolipid with disaccharide composed of glucose or mannose attached to fatty acids was supposed to represent a potential minimal ligand structure for Mincle [39,40]. The role of Mincle in innate immune response has been described in several fungi, but very few fungal ligands for this receptor were identified. Major surface glycoproteins of *Pneumocystis carinii* are recognized by Mincle through its short mannosylated chains [41], as well as mannosylated fatty acids in *Malassezia furfur* [42] and steryl 6-*O*-acyl-α-D-mannosides in *C. albicans* [43]. In contrast to the nature of these ligands, our results showed that both α-glucan polysaccharides from *L. prolificans* and *S. boydii* were recognized by Mincle. Deficiency in Mincle correlates with a delayed engulfment of fungal cells and impacts the phagocytosis dynamics [44], so by means of Mincle engagement, α-glucans may have a role in fungal phagocytosis. The absence of α-glucans binding to dectin-2 and dectin-3, however, is not surprising since these C-type lectin receptors recognize α-mannans on fungal surfaces, with dectin-2 presenting enhanced affinity for α-(1–2) or α-(1–4) Man units, especially in internal position [45,46].

Little is known about whether α-glucans can also modulate host defense by mechanisms mediated by dectin-1, the fungal β-glucan receptor. Interestingly, in our study, Dectin-1 recognized *L. prolificans* α-glucan, but the same was not observed for *S. boydii* glycogen-like α-glucan. Another glycogen-like highly branched α-glucan obtained from maitake mushroom (*Grifola frondosa*), however, has been shown to induce DC maturation by a Dectin-1-dependent pathway and not by TLR2, and Mincle, showing different binding properties from *S. boydii* α-glucan [47]. It has been reported that glucans variable immunomodulatory effects may depend on the degree of ramification [7,48] which can justify the differences between PRRs’ recognition of α-(1→4)-glucans from *L. prolificans* and *S. boydii*. 

It is noteworthy that cooperative interaction between Mincle and Dectin-1 or Mincle and TLR2 for the recognition of *L. prolificans* and *S. boydii* α-glucans, respectively, may influence the distinct immunostimulatory effects of these polysaccharides. The importance of co-stimulation of Mincle and TLR2 in TNF-α production and clearance of *Fonsecaea pedrosoi* was reported in [49], but nevertheless, the engagement of Mincle opposes Dectin-1-induced IL-12 production for antifungal TH1 immune responses [50]. Presumably, in our study, such differential sensing by TLR2 and Dectin-1 in synergy with Mincle of *L. prolificans* and *S. boydii* α-glucans contributes to their different TNF-α stimulatory capacity.

In conclusion, we described an α-(1→4)-glucan involved in *L. prolificans* phagocytosis and recognized by Dectin-1 and Mincle. We attempted to show that, probably due to the absence of branching, α-glucan from *L. prolificans* and that from the related species, *S. boydii*, are differently sensed by PRRs. 

## Figures and Tables

**Figure 1 jof-09-00291-f001:**
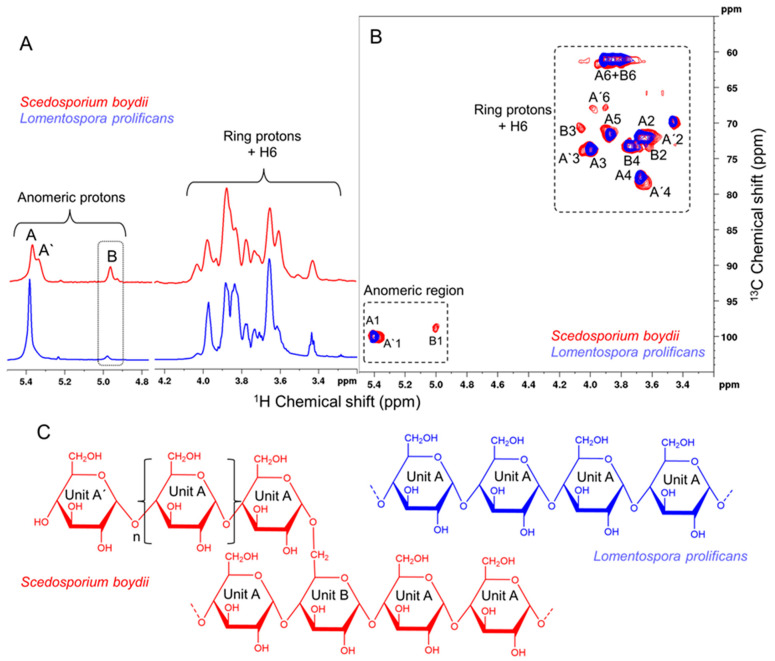
Structural determination of the glucans from *Scedosporium boydii* and *Lomentospora prolificans*: (**A**) 1D RMN 1H, (**B**) 2D 1H-13C HSQC, and (**C**) structural representation of glucans.

**Figure 2 jof-09-00291-f002:**
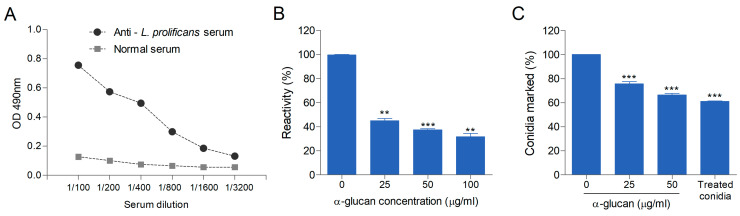
*Lomentospora prolificans* α-glucans is detected on conidia surface. (**A**). Reactivity of *L. prolificans* α-glucans with rabbit immune serum against whole *L. prolificans* cells (anti-*L. prolificans* serum) and with rabbit pre-immune serum (normal serum) at serial dilutions (1/100–1/3200) by ELISA. (**B**). Immune serum (dilution 1/400) bound to *L. prolificans* conidia, but pre-incubation with soluble α-glucan at 25, 50, and 100 µg/mL inhibited the reactivity measured by ELISA. (**C**). Intact conidia were incubated with rabbit immune serum against whole *L. prolificans* cells (dilution 1:400) or rabbit immune serum pre-incubated with different concentrations of soluble α-glucan (25 and 50 µg/mL) and analyzed by flow cytometry. Conidia treated with α-amyloglucosidase, which cleave off the →4)- αGlc (1→ units, were also incubated with rabbit immune serum against whole *L. prolificans* cells (dilution 1:400). Asterisks denote values significantly different from control (** *p* < 0.01; *** *p* < 0.001).

**Figure 3 jof-09-00291-f003:**
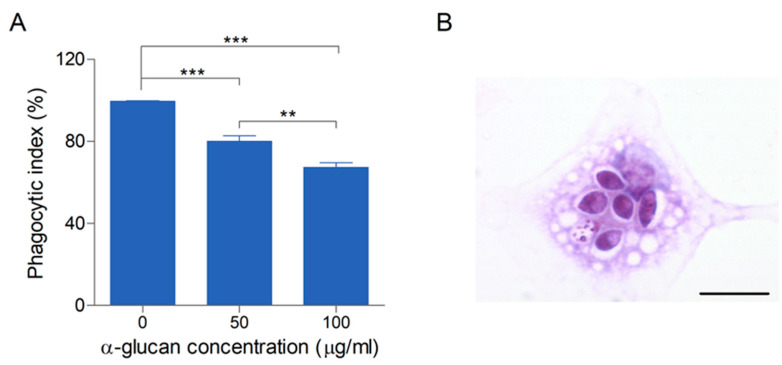
Phagocytosis inhibition assay between *L. prolificans* conidia and peritoneal macrophages by increasing concentrations of α-glucan. (**A**). The macrophages were pre-treated or not (control) for 30 min with two different concentrations of α-glucans (50 and 100 μg/mL), before interacting with conidial cells for 2 h. (**B**). Light micrograph of the interaction between *L. prolificans* conidia and peritoneal macrophages after 2 h. The phagocytic index values represent the mean ± S.D. of three independent experiments performed in triplicate. Bar: 20 µm. Asterisks denote values significantly different from control: ** *p* < 0.01 *** *p* < 0.001.

**Figure 4 jof-09-00291-f004:**
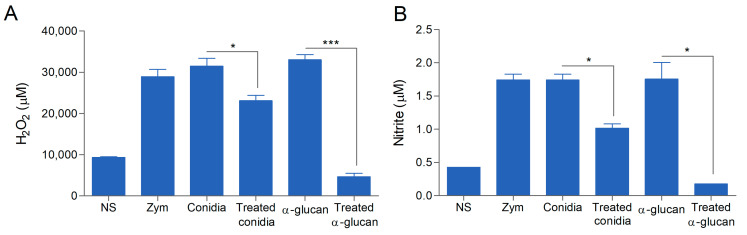
Involvement of *L. prolificans* α-glucan in the activation of oxidative burst from peritoneal macrophages. Macrophages were incubated in the presence of conidia treated or not with α-amyloglucosidase and with α-glucan (200 µg/mL) treated or not with α-amyloglucosidase at 37 °C for 18 h for quantification of H_2_O_2_ (**A**) and for 2 h for quantification of NO (**B**). Zymosan was used as a positive control (Zym). NS, non-stimulated macrophages. Data are the mean ± SEM of three independent experiments performed in triplicate. * *p* < 0.005 *** *p* < 0.0001.

**Figure 5 jof-09-00291-f005:**
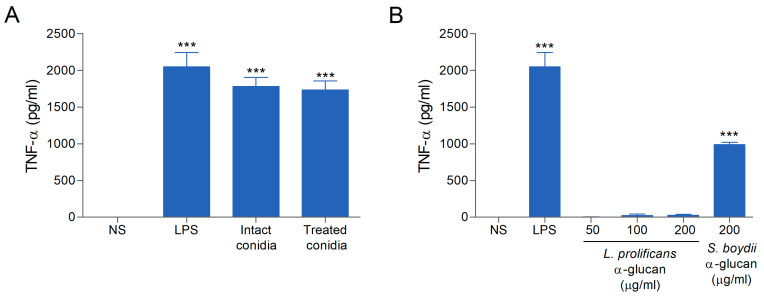
Induction of cytokine secretion by peritoneal macrophages. (**A**). Macrophages stimulated with *L. prolificans* conidia (treated or not with α-amyloglucosidase). (**B**). Macrophages stimulated with *L. prolificans* α-glucan at different concentrations (50, 100, and 200 µg/mL) and *S. boydii* α-glucan (200 µg/mL). After 18 h of incubation at 37 °C, culture supernatant was collected, and the concentration of TNF-α was determined by ELISA. NS, non-stimulated macrophages; LPS was used as positive control. Data are the mean ± SEM of three independent experiments performed in triplicate. *** *p* < 0.0001.

**Figure 6 jof-09-00291-f006:**
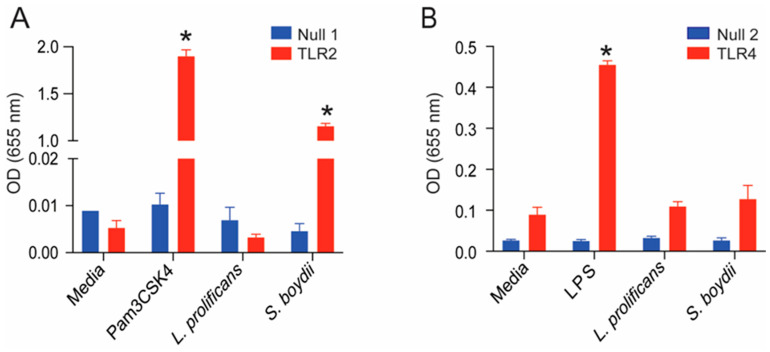
TLR2 recognizes *S. boydii* α-glucans but not those from *L. prolificans.* (**A**). HEK-Blue™ hTLR2 and HEK-Blue™ Null1 (control) were incubated with *L. prolificans* or *S. boydii* α-glucan. (**B**). HEK-Blue™ hTLR4 and HEK-Blue™ Null2 (control) were incubated with α-glucan from *L. prolificans* or *S. boydii* mycelia. LPS and Pam3CSK4 were used as positive control for TLR4 and TLR2, respectively. NF-kB SEAP activity was assessed using QUANTI-Blue™ and reading the OD at 655 nm. * *p* < 0.0001 vs. media control. Data representative of three independent experiments.

**Figure 7 jof-09-00291-f007:**
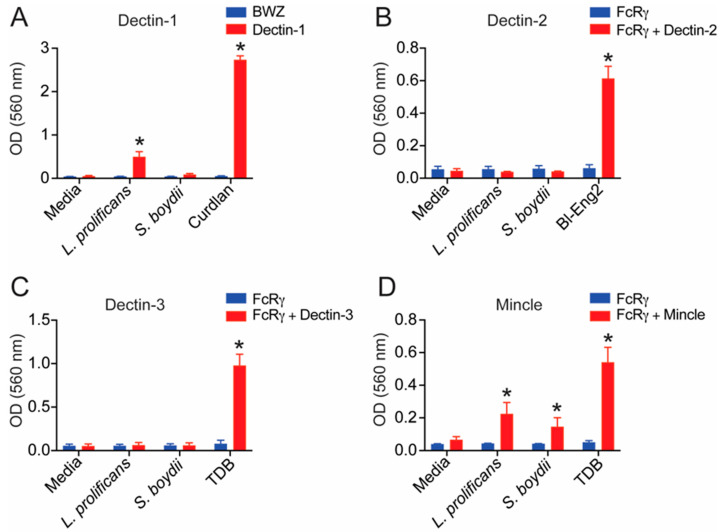
*Lomentospora prolificans* α-glucan is recognized by Dectin-1 and Mincle, but not by Dectin-2 or Dectin-3. BWZ or B3Z reporter cells expressing FcRγ plus Dectin-1 (**A**), Dectin-2 (**B**), Dectin-3 (**C**), or Mincle (**D**) were incubated with *L. prolificans* or *S. boydii* α-glucan. After 18 h of incubation, the β-galactosidase activity was measured by spectrophotometry. Curdlan (Dectin-1), Bl-Eng2 glycoprotein (Dectin-2), and TDB (Dectin-3 and Mincle) were used as positive control. * *p* < 0.001 vs. media control. Data representative of three independent experiments.

**Table 1 jof-09-00291-t001:** 1H and 13C Chemical shift (ppm) of the glucans from *S. boydii* and *L. prolificans*.

Atoms	→4)-αGlc(1→ (Unit A)	αGlc(1→4) (Unit A′)	→4,6)-αGlc(1→4) (Unit B)
** *S. boydii* ** ** ^a^ **			
H/C1	5.39/100.5	5.38/100.1	4.98/99.2
H/C2	3.66/72.2	3.66/72.3	3.61/ND
H/C3	3.96/73.9	3.97/74.0	4.00/ND
H/C4	3.66/78.0	3.64/78.0	3.65/78.8
H/C5	3.88/71.9	3.85/72.0	3.89/ND
H/C6	3.87/61.3	3.90/61.3	ND/ND
** *S. boydii* ** ** ^b^ **			
H/C1	5.38/100.1	5.34/100.1	4.97/98.9
H/C2	3.64/71.8	3.60/71.8	3.59/73.4
H/C3	3.97/73.5	4.02/73.6	4.03/70.6
H/C4	3.64/77.6	3.62/77.0	3.66/73.4
H/C5	3.85/71.7	3.88/70.6	ND
H/C6	3.87–3.77/60.9	3.96–3.87/67.6	3.87–3.77/60.9
** *L. prolificans* ^b^ **			
H/C1	5.38/100.1	ND	ND
H/C2	3.64/71.8	ND	ND
H/C3	3.97/73.5	ND	ND
H/C4	3.64/77.6	ND	ND
H/C5	3.85/71.7	ND	ND
H/C6	3.87–3.77/60.9	ND	ND

**^a^** Bittencourt, 2006 [7]. **^b^** HSQC this work, Figure 2A.

## Data Availability

Not applicable.

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
