# Peer review of "An Alpha-Glucan from Lomentospora prolificans Mediates Fungal–Host Interaction Signaling through Dectin-1 and Mincle"

_jof, 2023, doi:10.3390/jof9030291_

Round 1

Reviewer 1 Report

The manuscript communicates a relevant dataset that the specialized fungal immune sensing community will receive well. My only criticism is related to the localization of glucan on the cell surface. The evidence is weak and requires further confirmation. The major flaw of the experimental design is the polyclonal nature of the antibodies generated. Perhaps if cell wall preparations are used instead the experimental setting could be cleaner. Alternatively, the authors may attempt to purify by affinity chromatography the anti-glucan antibodies. Currently, there is no strong evidence to support this glucan is on the cell surface. In addition, how did the authors exclude the possibility that may also be localized in the inner part of the wall?

As a minor point, it would be interesting to analyze the genomic information of this organism and S. boydii, if available, and to draw working models to explain what genes may be responsible for the glucan branching in S. boydii.

Author Response

Reviewer 1

The manuscript communicates a relevant dataset that the specialized fungal immune sensing community will receive well. My only criticism is related to the localization of glucan on the cell surface. The evidence is weak and requires further confirmation. The major flaw of the experimental design is the polyclonal nature of the antibodies generated. Perhaps if cell wall preparations are used instead the experimental setting could be cleaner. Alternatively, the authors may attempt to purify by affinity chromatography the anti-glucan antibodies. Currently, there is no strong evidence to support this glucan is on the cell surface. In addition, how did the authors exclude the possibility that may also be localized in the inner part of the wall?

ANSWER: New experiments were made to reinforce the evidence that L. prolificans α-glucan is detected on cell surface and this result was re-written (Lines 268-282). Considering the results from Figures 2, which show that α-glucan is recognized by immune serum, and Figure 3, which show that it influences fungal phagocytosis, we believe that α-glucan is accessible on fungal surface. Otherwise, the macrophage receptors would not recognize the molecule. Moreover, a description of ELISA methodology was inserted in “Materials and Methods” section (Lines 127-142), and the use of α-amyloglucosidase enzyme was explained in “Materials and Methods” (Lines 116-117) and “Results” (Lines 280-281) sections.

As a minor point, it would be interesting to analyze the genomic information of this organism and S. boydii, if available, and to draw working models to explain what genes may be responsible for the glucan branching in S. boydii.

ANSWER: Genomic information of Scedosporium species is scarce and only the gene responsible for beta-glucan synthase is described so far. Nothing is known about the alpha-glucan. It is a very interesting idea, but would take a long time to perform a genetic model in Scedosporium. So, we will take it into consideration for further studies.

Reviewer 2 Report

This manuscript is relatively well presented and well written. Nevertheless, some corrections and additions would facilitate the understanding of certain sections.

Line 26 : Why "TNFalpha" ? It is a mistake I think

Line 70-71 : a space is missing

Line 211-212 : a word is missing in this sentence

Line 247, 265, 285 : why "alpha" in the title ?

§ 3.2 : this paragraph is a bit difficult to follow as it stands as well as Figure 2. Is it possible to add a figure to explain the protocol? Moreover you justify the use of the enzyme amyloglucosidase only in the discussion, without saying before that it fragments glucan. This is an important point to make beforehand. I did not understand your conclusion from the results presented in figure 2. this part needs to be revised in my opinion.

You do not use Figure 3B to support your point.

Figure 4 : What is "MO" ? and "Zym" ? Abbreviations are missing in the legend.

Figure 4-5 : I suggest replacing MO with non-stimulated "NS".

Line 325 : fashion by "manner" 

Also, how did you choose the different concentrations of glucan in your protocols? is it a physiological concentration, and observed during infections?

Author Response

Reviewer 2

Line 26: Why "TNFalpha"? It is a mistake I think

ANSWER: The sentence in line 26 was corrected: “Its recognition by macrophages is mediated by receptors that includes Dectin-1 and Mincle, but not TLR2 and TLR4.”

Line 70-71: a space is missing

ANSWER: A space was added (Line 70-71).

Line 211-212: a word is missing in this sentence

ANSWER: The sentence was corrected: “The signal B in the 1H NMR spectrum of the α-glucan from L. prolificans is much lower than that of S. boydii and, consequently, the absence of 5.33 ppm signal (A´).” (Line 233).

Line 247, 265, 285: why "alpha" in the title?

ANSWER: “α-glucan” in the title was corrected (Lines 268, 235, 346).

  • 3.2: this paragraph is a bit difficult to follow as it stands as well as Figure 2. Is it possible to add a figure to explain the protocol? Moreover, you justify the use of the enzyme amyloglucosidase only in the discussion, without saying before that it fragments glucan. This is an important point to make beforehand. I did not understand your conclusion from the results presented in figure 2. this part needs to be revised in my opinion.

ANSWER: The paragraph referring to the result showed in Figure 2 was re-written and new experiments were made to reinforce the evidence that L. prolificans α-glucan is detected on cell surface (Lines 269-282). Moreover, a description of ELISA methodology was inserted in “Materials and Methods” section (Lines 127-142), and the use of the α-amyloglucosidase enzyme was explained in “Materials and Methods” (Lines 116-117) and “Results” (Lines 280-281) sections.

You do not use Figure 3B to support your point.

ANSWER: This sentence was inserted in the result description to mention Figure 3B in the results: “Figure 2B demonstrate the interaction between L. prolificans conidia and peritoneal macrophages after 2 h.” (Lines 314-315)

Figure 4: What is "MO"? and "Zym"? Abbreviations are missing in the legend.

ANSWER: “MO” was changed to “NS”. The abbreviations of NS and “Zym” were added in the legend of Figure 4.

Figure 4-5: I suggest replacing MO with non-stimulated "NS".

ANSWER: “MO” was replaced by “NS”.

Line 325: fashion by "manner"

ANSWER: “fashion” was replaced by “manner” (Line 404).

Also, how did you choose the different concentrations of glucan in your protocols? is it a physiological concentration, and observed during infections?

ANSWER:  The different concentrations of L. prolificans α-glucan were chosen according to our previous work with α-glucan from Scedosporium boydii (Bittencourt et al. 2006), in which it was also tested at 25, 50 and 100 µg/ml. We also evaluated the cytotoxicity of L. prolificans α-glucan, since it presents structural differences, and the result demonstrates that L. prolificans α-glucan did not show any toxicity against peritoneal macrophages up to 200µg/ml, the maximum concentration tested.

Regarding the physiological concentration of α-glucan on scedosporiosis and lomentosporiosis, it is still unknown. For filamentous and dimorphic fungal infection, such as Histoplasma capsulatum and Aspergillus fumigatus, which present either β- and α-glucan, these polysaccharides can be detected in body fluids and serum, but the antigen level is variable (Wagener et al. 2020; Plaza et al. 2020).

Bittencourt VC, Figueiredo RT, da Silva RB, Mourão-Sá DS, Fernandez PL, Sassaki GL, Mulloy B, Bozza MT, Barreto-Bergter E. An alpha-glucan of Pseudallescheria boydii is involved in fungal phagocytosis and Toll-like receptor activation. J Biol Chem. 2006 Aug 11;281(32):22614-23. doi: 10.1074/jbc.M511417200. Epub 2006 Jun 9. PMID: 16766532.

Wagener, J., Striegler, K., Wagener, N. (2020). α- and β-1,3-Glucan Synthesis and Remodeling. In: Latgé, JP. (eds) The Fungal Cell Wall. Current Topics in Microbiology and Immunology, vol 425. Springer, Cham. https://doi.org/10.1007/82_2020_200

Plaza V, Silva-Moreno E, Castillo L. Breakpoint: Cell Wall and Glycoproteins and their Crucial Role in the Phytopathogenic Fungi Infection. Curr Protein Pept Sci. 2020;21(3):227-244. doi: 10.2174/1389203720666190906165111. PMID: 31490745.

Round 2

Reviewer 1 Report

Thanks for addressing my concerns.